# Highly Hydrophobic Cotton Fabrics Modified by Poly(methylhydrogen)siloxane and Fluorinated Olefin: Characterization and Applications

**DOI:** 10.3390/polym12040833

**Published:** 2020-04-06

**Authors:** Huiping Lin, Qingjian Hu, Tianyu Liao, Xinxiang Zhang, Wenbin Yang, Shuang Cai

**Affiliations:** 1College of Materials Engineering, Fujian Agriculture and Forestry University, Fuzhou 350108, China; 18450077642@163.com (H.L.); hqj0215@163.com (Q.H.); lty1214745470@163.com (T.L.); xxzhang0106@163.com (X.Z.); 2College of Chemical Engineering, Hubei University of Arts and Science, Xiangyang 441053, China

**Keywords:** cotton fabric, hydrophobic, poly(methylhydrogen)siloxane, fluorinated olefin, oil–water separation

## Abstract

Highly hydrophobic cotton fabrics were obtained with poly(methylhydrogen)siloxane (PMHS) and a further fluorinated olefin modification. The chemical structures and microstructures of PMHS-modified cotton fabrics were characterized, and application of the resultant cotton fabrics in stain resistance and oil–water separation was demonstrated. PMHS chains with very low surface energy were grafted onto cotton fabric by the dehydrogenation reaction between –Si–H of PMHS and –OH groups of cotton fabric at room temperature. The water contact angle of PMHS-modified cotton fabric was 141.7°, which provided the modified cotton fabric with good stain resistance to waterborne pollutants. The separation efficiency of diesel from water was higher than 92% for 20 repeatable separation cycles. A further improvement in stain resistance to oil was also demonstrated by a further addition reaction of 1H,1H,2H-perfluoro-1-decene with PMHS-modified cotton fabric.

## 1. Introduction

Cotton fabrics are widely applied as textiles because they are soft, comfortable, breathable, and do not irritate human skin [1]. Moreover, they are also used as absorbent materials due to cotton’s microscale porous structure [2]. However, cotton fabrics possess two arresting disadvantages: unsatisfactory stain resistance to liquids and poor selectivity in liquid absorption [3]. Normal cotton fabrics can be easily stained by juice, pigment, and blood, and they are unable to selectively separate oils from the oil/water mixture. Therefore, it is highly desirable to improve the stain resistance of cotton fabrics and afford them unique oil/water separation ability.

Fabricating a superhydrophobic surface on cotton fabrics has been extensively studied, which has been demonstrated as an effective method in the functionalization of cotton fabrics for self-cleaning [4,5] and oil/water separation [6,7,8,9,10]. The superhydrophobic surface on cotton fabrics could be realized by various techniques, including the CVD (chemical vapor deposition) method [11,12,13], sol–gel process [3,14,15], solution immersion [16,17], spray deposition [18,19,20], and graft polymerization [1,12,21,22,23,24,25]. Two essential factors for the fabrication of a superhydrophobic surface on cotton fabrics are (i) rough surface and (ii) low surface energy. To build a rough surface on cotton fabrics, inorganic nanoparticles such as ZnO [26], TiO_2_ [6,7,15,27], and SiO_2_ [2,14,28] were employed. Since the inorganic nanoparticles were generally very hydrophilic, a further hydrophobic modification process should be carried out to lower the surface energy of the resultant rough surface. The modifiers could be classified as stearic acid [29], n-octadecylthiol [30], chlorosilane (octadecyltrichlorosilane [31]), siloxane (R’Si(OR)_3_, R=–CH_3_, –CH_2_CH_3_; R’=–(CH_2_)_n_CH_3_, or –CH_2_CH_2_(CF_2_)_n_CF_3_) [11,32,33,34,35,36], or polyhedral oligomeric silsesquioxane (POSS) [37,38,39]. Among them, fluorine-containing compounds, which are characterized by long fluoroalkyl chains of C8 or higher [35,36,40,41,42], have most commonly been utilized owing to the ultra-low surface energy. 

To date, the fabrication of a superhydrophobic surface on cotton fabrics has been extensively investigated. However, when taking practical applications into account, the reported approaches possess several disadvantages. First, fluorinated compounds are expensive [43]. Second, hierarchical structures fabricated on cotton fabrics possess poor stability because they commonly consist of inorganic nanoparticles, which were randomly stacked on the cotton fabrics. The bonding among the inorganic nanoparticles is quite poor. Although many works have attempted to improve the bonding of nanoparticles to cotton fabrics and also the bonding between individual nanoparticles by organic materials (such as polymers) [5], the organic/inorganic hybrid coatings on cotton fabrics cannot be thin enough to keep the original appearance (especially for colored cotton fabrics), softness, and wearing comfort of pristine cotton fabrics [1]. Third, most of the methods for the fabrication of superhydrophobic surfaces on cotton fabrics are too complicated to be used practically. 

Cotton fabrics possess inherent surface roughness on the microscale due to the textural structure and complicated weave pattern [39]. Therefore, highly hydrophobic cotton fabrics can be realized by a one-step hydrophobic modification of cotton fabrics with hydrophobic materials with very low surface energy [1,44]. In practical application, superhydrophobicity is not a necessary condition for the stain resistance and oil/water separation of cotton fabrics, and a highly hydrophobic property is enough. Actually, for application in stain resistance and oil/water separation, highly hydrophobic cotton fabrics modified by fluorine-free materials can overcome the aforementioned disadvantages of superhydrophobic cotton fabrics. 

The aim of this work is to fabricate highly hydrophobic cotton fabrics by the covalent grafting of poly(methylhydrogen)siloxane (PMHS) and a further addition of fluorinated olefin (1H,1H,2H-perfluoro-1-decene). In consideration of the molecular structure, PMHS contains amounts of –Si-H groups and –CH_3_ groups [45,46]. The former groups can react with the hydroxyl groups of cotton fabrics, and the latter groups afford PMHS chains very low surface energy. Therefore, PMHS was used to functionalize the cellulose-based substrates for water repellence and oil–water separation [47,48]. In this work, after modification, hydrophobic PMHS chains can be covalently bonded onto the cotton fabrics. PMHS is transparent [49] and PMHS chains possess good flexibility [50]; therefore, PMHS modification will probably not affect the original appearance, softness, and wearing comfort of cotton fabrics. To further improve the resistance of cotton fabric to oil, a further addition reaction was applied to modify PMHS-modified cotton fabric with 1H,1H,2H-perfluoro-1-decene. Pristine and modified cotton fabrics were characterized systematically, and their application in stain resistance and oil/water separation was well demonstrated. 

## 2. Experimental Sections

### 2.1. Materials

PMHS with a molecular weight of about 1600 and a content of active hydrogen (Si-H) of 1.5% and Karstedt catalyst (platinum-1,3-divinyl-1,1,3,3-tetramethyldisiloxane) with a Pt content of 2000 ppm were purchased from the Chenguang Research Institute of Chemical Industry (Chengdu, China). 1H,1H,2H-Perfluoro-1-decene, hexane, and methylene blue (MB) were obtained from ALADDIN Reagent Co., Ltd (Shanghai, China). The cotton fabrics were obtained from a local fabric store (60 ends/cm, 30 picks/cm, 0.42 mm thickness, 120 g/m^2^ weight, and 35.2 m^2^/g specific surface area). 

### 2.2. Surface Modification of Cotton Fabrics

#### 2.2.1. PMHS Modification

The modifier solutions with various PMHS concentration of 0.05 wt %, 0.10 wt %, 0.50 wt %, 1.00 wt %, 5.00 wt %, 10.00 wt %, and 20.00 wt %, respectively, were obtained by magnetic stirring of PMHS, 50 mL hexane, and 3 drops of Karstedt catalyst (about 0.15 mL) at room temperature for about 5 min. The hydrophobic modification of cotton fabrics was carried out by a very simple process. Before PMHS modification, cotton samples (4.0 cm × 4.0 cm) were ultrasonically rinsed several times with ethanol and deionized water and then dried at 100 °C for 6 h. Then, the cotton fabrics were immersed into the modifier solutions at room temperature for about 5 min and then withdrawn and maintained at room temperature for the evaporation of hexane. During modification, PMHS reacts with OH groups of cellulose, resulting in a covalent linkage of hydrophobic PMHS chains onto cotton fabrics (Scheme 1). 

#### 2.2.2. Fluorinated Olefin Modification

Cotton fabric modified by a modifier solution with a PMHS concentration of 10.0% was further modified by immersing into the 5 wt % 1H,1H,2H-perfluoro-1-decene/hexane solution for 5 min at room temperature and then withdrawing and maintaining it at room temperature for the evaporation of hexane. To accelerate the evaporation of hexane, the modified fabric can be heated at 60 °C for 1 h. During modification, an addition reaction occurred between the vinyl groups of 1H,1H,2H-perfluoro-1-decene and the residual –Si-H groups of the PMHS-modified cotton fabric, and therefore, long fluoroalkyl chains with ultra-low surface energy were grafted onto the surface of the cotton fabric (Scheme 2). 

### 2.3. Characterization

Infrared absorption spectra of pristine and PMHS-modified cotton fabrics were analyzed by FTIR (Fourier transform infrared spectroscopy) (Bruker Tensor 27, Bruker Optik, Ettlingen, Germany) using the attenuated total reflectance (ATR) method (ranging from 4000 to 400 cm^−1^, 32 scans, resolution of 0.1 cm^−1^). For the ATR method, it is very important to obtain good contact between the crystal and the cotton fabrics, which was achieved by pressing the cotton fabrics down on the ZnSe crystal. The XPS (X-ray photoelectron spectroscopy) spectra of the samples were obtained by using a Thermo SCIENTIFIC ESCALAB 250Xi (Thermo Fisher Scientific, Pittsburgh, PA, USA) with Al X-radiation (Kα, h*v* = 1486.8 eV). Two different spots (spot size = 400 μm) were analyzed per sample at take-off angle 90°. The power was 150 W (15 kV and 10 mA), and the vacuum of the sample chamber was 2 × 10^−9^ mbar. The survey spectra were acquired with a pass energy and resolution of 30 eV and 1 eV, respectively. An electron flood gun was used to compensate for the charges on the surface. Note that the samples for FTIR and XPS characterization were washed with hexane three times to remove PMHS, which did not react with the cotton fabrics. For the test of water contact angle (WCA) values, rectangular samples (40 × 20 mm^2^) of the cotton fabrics were cut and pasted onto glass slide with double-face tape. The WCA of cotton fabric was measured on a commercial contact angle meter (HARKE-SPCA-1, Beijing, China) at room temperature with 5 μL water droplets. The surface morphology and microstructure of the cotton fabrics were characterized using a scanning electron microscope (SEM, Z500, ZEISS, Jena, Germany) operating at an acceleration voltage of 20 kV in combination with energy-dispersive X-ray spectroscopy (EDS, Genesis, EDAX Inc., Mahwah, NJ, USA). Before characterization, the samples were coated with a layer of platinum. 3. Results and Discussion

## 3. Results and Discussion

### 3.1. Chemical Structure Analysis

The modification of cotton fabrics by PMHS is based on the dehydrogenation reaction between the –Si-H groups of PMHS and the –OH groups of cotton fabrics. Figure 1 shows the FTIR-ATR spectra of pristine and PMHS-modified cotton fabrics. For the pristine cotton, absorption bands at 2916 cm^−1^ were attributed to the –CH asymmetric and symmetric stretching vibrations of methylene groups of cellulose [51]. Absorption bands at 1161, 1108, 1056, and 1032 cm^−1^ were attributed to C–O–C vibrations in the cellulose intra- and intermolecular structure [51,52]. For PMHS-modified cotton fabric, the grafting of PMHS chains on cotton fabrics is demonstrated by the appearance of bands from 2965, 1260, 835, and 765 cm^−1^ that correspond to the vibration of the Si-CH_3_ group [53,54,55,56,57] of the PMHS chain and by an absorption band located at 2165 cm^−1^ that corresponds to the Si-H asymmetric stretch mode of the PMHS chain [58]. The obvious appearance of an –Si-H absorption band was attributed to the steric effect of plentiful –CH_3_ groups on the PMHS chains. In addition, the dehydrogenation reaction between PMHS and cotton fabrics is also proven by the significant decrease in the –OH absorption band at 3337 cm^−1^ [59]. Since the samples for FTIR characterization have been washed by hexane three times, it can be concluded that the PMHS chains were covalently grafted onto the cotton fabrics. 

As shown in Figure 2, XPS spectra show the composition changes of cotton fabrics before and after modification. As shown in Figure 2a, C, O, and very few Si elements were detected at the binding energy of 284.8, 531.4, and 102.0 eV on the surface of pristine cotton fabrics [60,61]. As shown in Figure 2b, Si content was significantly increased from 4.0% to 32.0%, while C and O content decreased obviously from 69.1% and 26.9% to 35.2% and 32.85, respectively. This is in good agreement with the results of FTIR, in which PMHS chains were concluded to be grafted onto the surface of the cotton fabric. As shown in the inserted image in Figure 1b, after PMHS modification, there is probably a layer of PMHS fabricated on the cotton fabric, and therefore the XPS spectrum of PMHS-modified fabric will be similar to that of PMHS, whose C/O/Si atomic ratio is about 1:1:1. 

### 3.2. Microstructure Characterization

Figure 3a,b,d,e show the SEM images of pristine cotton fabric and PMHS-modified cotton fabric. At low magnification, SEM images (Figure 3a,d) indicated that no significant morphology change was found between the unmodified and PMHS-modified cotton fabrics. This is because there is only a thin PMHS layer formed on surface of the cotton fabric. However, at higher magnification, some delicate differences were observed as shown in Figure 3b,e. The gaps between adjacent fibers in unmodified cotton fabric were clear, but they were replaced by PMHS for the modified cotton fabric. As at higher magnification, the surface of PMHS-modified fibers was smoother due to the covering of the PMHS layer. 

### 3.3. Hydrophobicity and Water Resistance

Figure 4 shows the effect of PMHS concentration in modifier solution on the water contact angle and water absorption of cotton fabrics. As shown in Figure 4b,c, the water droplets are difficult to be placed on the surface of cotton fabrics because they are highly hydrophobic. It is hard to record their WCA values. In this work, the WCA was tested with the needle inserted into the water droplet. It can be deduced that the recorded WCA values will be smaller than the actual values. In addition, the measurement of WCA was experienced because the estimation of the baseline for the determination of the WCA is much harder due to the existence of protruding fibers in the woven structure [62,63]. Regardless of these, as shown in Figure 4, it can be seen that PMHS modification significantly improves the hydrophobicity of cotton fabrics. For the pristine fabric, the water droplet was absorbed quickly due to the very hydrophilic property of cellulose and wicking of water, and therefore, its WCA value is about 0°. The PMHS concentration of the modifier solution has little effect on the WCA value. This is probably because of the fast fabrication of the PMHS layer onto the cotton fabric. One of the advantages of our modifier method is that the grafting of PMHS chains can be accomplished very fast at room temperature. With Karstedt catalyst, the dehydrogenation between –Si-H of PMHS and –OH of cellulose is ultra-fast. Therefore, as soon as the cotton fabric was immerged into the modifier solution, a thin PMHS layer was formed on its surface. The sliding angles of the water droplets on PMHS-modified cotton fabrics were also tested, which were all lower than 10°. This is also demonstrated vividly in Appendix A, in which the water droplets (dyed by MB) immediately rolled of the 10% PMHS-modified cotton fabric easily. Therefore, in this work, it is difficult to conclude that PMHS-modified cotton fabrics were superhydrophobic, because the static WCA was tested to be lowered than 150°. It is well known that PMHS possesses very low surface energy, and hence the covalent graft of PMHS on will stop cotton fabric absorbing water from water. Water absorption was applied to reveal the hygroscopicity of cotton fabric, which was calculated as follows:(1)$$\mathrm{Water}\;\mathrm{absorption}=\frac{m_{1}-m_{0}}{m_{0}}$$
where m_0_ and m_1_ were the mass of cotton fabrics before and after water immersion for 24 h. As shown in Figure 4, the water absorption of unmodified cotton fabric is higher than 170%, while those of the modified ones are obviously decreased to about 30%, indicating good water resistance for the modified cotton fabrics. As shown in Figure 4d, the unmodified cotton fabric immediately absorbed water and sank into the water. However, for the modified one, it floated on the water surface because it cannot be wetted. After being forcibly immersed into water, a mirror-like surface was observed on the cotton fabric, owing to the reflection of light by the air bubbles trapped on the surface of the cotton fabric [28]. Furthermore, PMHS concentration has little effect on the wettability of modified cotton fabrics. This is because PMHS is of very low surface energy, and the dehydrogenation between –Si-H of PMHS and –OH of cellulose in cotton fabric is ultra-fast [58]. Therefore, a very hydrophobic PMHS layer will form on the surface of cotton fabrics as they were immersed into PMHS modifier solutions.

### 3.4. Application in Stain-Resistance

In order to demonstrate the stain resistance of PMHS-modified cotton fabric to waterborne pollutants, aqueous MB droplets were dripped onto the surface of cotton fabrics. As shown in Figure 5a, the MB droplets spread easily on the surface of pristine cotton fabrics, and a distinct mark was observed after absorbing aqueous MB droplets by a tissue. This indicated that the pristine cotton fabric possessed poor stain resistance. As shown in Figure 5b–h, the modification of cotton fabric with PMHS significantly improved the water repellence, as the aqueous MB droplets displayed good spherical shapes. Since aqueous MB droplets could not spread on the surface of PMHS-modified cotton fabrics, there were inconspicuous marks on surface of the cotton fabrics modified by modifier solutions with PMHS concentration lower than 1%. At the PMHS concentration higher than 5%, no macroscopic mark was observed. Moreover, as shown in Appendix A, the water droplets (dyed by MB) immediately rolled of the 10% PMHS-modified cotton fabric easily. This demonstrated convincingly that PMHS-modified cotton fabrics possessed good stain resistance to waterborne pollutants. 

In addition to stain resistance to waterborne pollutants, PMHS-modified cotton fabrics can gain stain resistance to oil by a further addition reaction with 1H,1H,2H-perfluoro-1-decene. As shown in the FTIR spectra (Figure 1), there was a significant absorption band at 2165 cm^−1^ for PMHS-modified cotton, indicating that PMHS-modified cotton fabric was rich in –Si–H bonds. With Karstedt catalyst, the addition reaction between –Si–H of PMHS-modified cotton fabric and vinyl groups of 1H,1H,2H-perfluoro-1-decene occurs easily at room temperature, and therefore, the long fluoroalkyl chains can be grafted onto the cotton fabrics by a simple immerging method. As shown in XPS spectra (Figure 2e), there was an addition peak at 689.19 eV attributed to the F element [64]. As shown in Figure 6, as the oil droplet (about 0.05 mL) was dripped onto pristine cotton fabric, the oil spread easily, and quite a big area of cotton fabric was stained by oil. PMHS modification of cotton fabric had no benefit regarding the stain resistance of cotton fabric to oil, as the oil droplet also spread quickly on surface of the PMHS-modified cotton fabric. This is because PMHS chains were abundant in –CH_3_ groups, which leads to an oleophilic characteristic for PMHS-modified cotton fabric. Long fluoroalkyl chains are well known for their ultra-low surface energy and oil resistance. As shown in Figure 6c, the oil droplet was stable, and no spread was observed on 1H,1H,2H-perfluoro-1-decene-modified cotton fabric. This provides an idea for the preparation of cotton fabric with stain resistance to both water and oil, and systematic research can be carried out based on this work. 

### 3.5. Application in Oil–Water Separation

The application of PMHS-modified cotton fabric in oil–water separation was also demonstrated. As shown in Figure 7, because the density of diesel was smaller than water, the separation apparatus should be tilted. To determine the separation efficiency, 10 mL of diesel and 10 mL of water were mixed and poured into the separation apparatus continuously. As shown in Figure 7b, the highly hydrophobic cotton fabrics can only absorb diesel but repel water, and as a result, only diesel flowed through the cotton fabrics and fell down to the conical flask. Figure 7a shows the change in separation efficiency as a function of separation cycles. After 20 repeatable separation cycles, the separation efficiency of diesel from water was demonstrated to be higher than 92%. This indicated good oil–water separation ability and a good recyclability of PMHS-modified cotton fabrics. 

## 4. Conclusions

In summary, highly hydrophobic cotton fabrics were fabric by PMHS modification. PMHS chains with very low surface energy were covalently bonded to the surface of cotton fabrics. PMHS modification did not significantly affect the microstructure of cotton fabric. After PMHS modification, the water contact angle of cotton fabric was significantly improved to 141.7°. Excellent water repellence afforded the resultant cotton fabric good stain resistance to waterborne pollutants. PMH-modified cotton fabrics also possessed good oil–water separation ability, with the separation efficiency higher than 92%. PMHS-modified cotton fabrics are abundant in –Si-H groups, and a further improvement in stain resistance to oil was also demonstrated by an additional reaction with 1H,1H,2H-perfluoro-1-decene.

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
