# Peer review of "Highly Hydrophobic Cotton Fabrics Modified by Poly(methylhydrogen)siloxane and Fluorinated Olefin: Characterization and Applications"

_polymers, 2020, doi:10.3390/polym12040833_

Round 1

Reviewer 1 Report

The paper presents the modification of the wettability properties of cotton fabrics by dehydrocoupling reaction with PMHS and hydrosilylation with a fluorinated olefin. The novelty/originality of the approach appears poor, PMHS was already used for this purpose, and modification of Si-based layers with hydrophobic chains was already reported for further tuning the surface energy.

The authors are required to carefully revise the manuscript according to the following comments and reporting additional experiments.

Introduction must include the use of PMHS and similar polymers (for modifying the surface properties of polymers and biopolymers as cellulose, for oil-water separation), reporting the appropriate references.

Experimental section must be improved, there are many lacking information

  • Section 2.1: Which is the MW of PMHS? Which are the features of cotton fabrics?
  • Section 2.2.1 : Did the authors perform any pre-treatment in order to remove the substances normally used on fabrics such as dressing? Which is the Karstedt’s catalyst (and not Kastredt as always reported in the text) concentration used for dehydrocoupling?
  • Section 2.2.2: the use of Karstedt’s catalyst for hydrosilylation is not reported, please add it with the employed concentration
  • Section 2.3: Add more details on FTIR (ATR crystal, range, scans, resolution); which is the take-off angle used in XPS analyses? Did you consider the effect of charging? Any charge referencing for assessing peaks position?

Results and discussion:

  • The FTIR spectrum of pristine cotton fabric is not described, in Figure 1 the signals at 1161, 1108, 1056 and 1032 cm-1 are highlighted but no comments in the text
  • Page 3, Line 124: the decrease of OH signal of cotton is pointed out, but the spectrum of PMHS-modified cotton does not show any signal of the substrate. Which is the PMHS layer thickness? considering the depth of sampling with usual ATR crystals, the layer should be thick but at page 5, line 152 the authors affirm that the layer is thin. Please, add thickness information and justify the spectral features
  • For completeness, the pristine PMHS spectrum should be added to Figure 1
  • Page 4: a) the assignment of the different components of cotton, PHMS-cotton and fluorinated-PMHS-cotton C 1s spectra must be reported and discussed in the text adding suitable references; b) please justify the energy values that appear different from those reported in the literature, in particular what about component C1 in figure 2b?
  • Page 4,Line 138-139: please show the C1s spectrum of PMHS in order to confirm the assignments, add references
  • Page 4, Line 134: 2b must be corrected in 2c
  • Page 4, Line 141 : 2(d and e) is 2(b and d)
  • Page 5: improve the caption of Figure 2
  • Page 6, lines 156-158: the sentence must be removed, no correlation with the text above
  • Section 3.3 : to improve the soundness of the results considering the comments made at page 6 lines 164-169, the wettability study of rough cotton surfaces must be completed with shedding angle measurements that should be discussed together with WCA data
  • Page 6, Lines 172-174: please explain why PMHS concentration does not have effect on wettability. Haw many measurements were done? Please report error bars on WCA and water absorption values in Figure 4.
  • Page 7 line 199: a video is mentioned, I didn’t find it in the submitted material
  • Figures 1 and 4 must be enlarged, Figure 4 (and caption) must be improved, too many information; consider to prepare fig4a and b with plot and pictures respectively .
  • At page 2 lines 66-68, the authors justify PMHS selection because it assures the maintenance of original features of cotton fabrics (flexibility,…). Accordingly, the evaluation of the coated cotton fabrics behavior and the study of coating mechanical resistance ( bending test, layer adhesion, ..) must be added.

Revise the conclusions in accordance with new experimental data and improved discussion

Author Response

Responses to comments of Reviewer1

Q1: Introduction must include the use of PMHS and similar polymers (for modifying the surface properties of polymers and biopolymers as cellulose, for oil-water separation), reporting the appropriate references.

Answer: Thanks for your suggestions. We have added the details of functionalization of biopolymers for oil-water separation and the relevant references in the new submitted manuscript.

Q2:Experimental section must be improved, there are many lacking information. (1) Section 2.1: Which is the MW of PMHS? Which are the features of cotton fabrics? (2) Section 2.2.1 : Did the authors perform any pre-treatment in order to remove the substances normally used on fabrics such as dressing? Which is the Karstedt’s catalyst (and not Kastredt as always reported in the text) concentration used for dehydrocoupling? (3) Section 2.2.2: the use of Karstedt’s catalyst for hydrosilylation is not reported, please add it with the employed concentration. (4) Section 2.3: Add more details on FTIR (ATR crystal, range, scans, resolution); which is the take-off angle used in XPS analyses? Did you consider the effect of charging? Any charge referencing for assessing peaks position?

Answer: We have revised the Experimental Section according to your suggestions. Thank you very much. (1) The MW of PMHS is about 1600, which had been added in the new submitted manuscript. (2) Sorry for no details about the cotton fabrics. We have added the details. (3) In our work, we added 3 drops (about 0.15 mL) Karstredt catalyst into 50 mL Hexane, and the addition of PMHS was calculated to afford the PMHS concentration of 0.05%, 0.10%, 0.50%, 1.00%, 5.00%, 10.00%, 20.00%. (4) We have added the details in the new submitted manuscript.  

Q3: The FTIR spectrum of pristine cotton fabric is not described, in Figure 1 the signals at 1161, 1108, 1056 and 1032 cm-1 are highlighted but no comments in the text.

Answer: Thanks for your careful review on this detail. These for signals are attributed to cellulose. After modification, they were covered by the signals of PMHS. We have added the discussion of FTIR before PMHS modification. Thanks again.

Q4: Page 3, Line 124: the decrease of OH signal of cotton is pointed out, but the spectrum of PMHS-modified cotton does not show any signal of the substrate. Which is the PMHS layer thickness? considering the depth of sampling with usual ATR crystals, the layer should be thick but at page 5, line 152 the authors affirm that the layer is thin. Please, add thickness information and justify the spectral features.

Answer: Actually, for PMHS modified cotton, it is difficult to present the PMHS signal in FTIR spectrum from ATR model. This is probably because the PMHS layer on surface of cotton fabric is quite thin. So, for FTIR characterization, the sample was immerged in the PMHS modifier solution for a quite long time (more than 0.5 h) to obtain a thicker layer for FTIR characterization. This operation is an exception only for FTIR characterization. For application, the cotton fabrics were immerged in the modifier solutions for 5 minutes as described in the Experimental Section.

Q5: For completeness, the pristine PMHS spectrum should be added to Figure 1

Answer: We are sorry that we cannot test the FTIR spectrum of pristine PMHS due to the coronavirus. Our campus is closed. We think the FTIR spectra have revealed the difference before and after modification, and the characteristic absorption bands of PMHS, e.g. -Si-CH3 and –Si-H, were recorded.

Q6: Page 4: a) the assignment of the different components of cotton, PHMS-cotton and fluorinated-PMHS-cotton C 1s spectra must be reported and discussed in the text adding suitable references; b) please justify the energy values that appear different from those reported in the literature, in particular what about component C1 in figure 2b?

Answer: There are indeed some problems in the deconvoluted C1s XPS spectrum of pristine cotton fabric. We cannot assign the C1S at 283.37 eV according to the references. Because the characterization of cotton fabric in our work is quite comprehensive, and we delete the Figures and discussion about deconvoluted C1s XPS spectra in the new submitted manuscript. Thank you very much.

Q7: Page 4,Line 138-139: please show the C1s spectrum of PMHS in order to confirm the assignments, add references

Answer: We are very sorry that we cannot carry out characterization due to the coronavirus. Also, in the new submitted manuscript, discussion about econvoluted C1s XPS spectra was deleted. We appreciate your understanding.

Q8: (1) Page 4, Line 134: 2b must be corrected in 2c; (2) Page 4, Line 141 : 2(d and e) is 2(b and d)

Answer: We have deleted Figure 2(b, d and f) in the new submitted manuscript.

Reviewer 2 Report

Dear Authors,

the manuscript entitled "Highly hydrophobic cotton fabrics modified by poly(methylhydrogen)siloxane: characterization and applications” by Huiping Lin, Qingjian Hu, Tianyu Liao, Xinxiang Zhang, Wenbin Yang and Shuang Cai investigates the use of poly(methylhydrogen)silixane (PMHS) to modify cotton fabrics and a second modification with 1H,1H,2H-Perfluoro-1-decene.

The objective of the article is clear, but has been much investigated with different approaches, the literature reported is not complete and more comparisons should be made with the results obtained in this research to evaluate the effectiveness of the proposed solution. In general the whole article is a little superficial from the scientific point of view, it should be embellished and deepened.

E.g. some articles quickly found by the reviewer but many others will exist: “Transformation of hydrophilic cotton fabrics into superhydrophobic surfaces for oil/water separation” ; “Robust and Durable Superhydrophobic Cotton Fabrics for Oil/Water Separation” ; “Fabrication of Durably Superhydrophobic Cotton Fabrics by Atmospheric Pressure Plasma Treatment with a Siloxane Precursor”; how is this article different, has superior or inferior properties, and why?

  • In the title and in the abstract the second modification with fluorine olefin is not mentioned. There is the description in the experimental part and then in the results it is sometimes treated, sometimes not. The text must be revised so that all the characterizations are present and all the comments and descriptions in all the parts regarding this second modification.
  • Whenever an outcome is defined, an appropriate reference must be cited. For example at each attribution of the IR.
  • Control of the English language used which sometimes needs to be improved for verbal forms and repetitions and check the references to figures that in some cases are missing or incorrect. Figure 1 (b) defined in the text where is it? The caption of figure 2 seems incorrect. Line 150, the references seem incorrect. Figures 3c and f, Figure 3 g, h, i are not commented on in the text.
  • To verify the chemical reaction and also to evaluate its quantity, why have not been made thermogravimetry on the treated materials?
  • In the section 3.5 the second treatment must be presented and compared with the first one. Furthermore, probably another index concerning the performance of the separation could be introduced. For example the time to separate the two phases.

Author Response

Response to Reviewer2:

Q1: The objective of the article is clear, but has been much investigated with different approaches, the literature reported is not complete and more comparisons should be made with the results obtained in this research to evaluate the effectiveness of the proposed solution. In general the whole article is a little superficial from the scientific point of view, it should be embellished and deepened. E.g. some articles quickly found by the reviewer but many others will exist: “Transformation of hydrophilic cotton fabrics into superhydrophobic surfaces for oil/water separation” ; “Robust and Durable Superhydrophobic Cotton Fabrics for Oil/Water Separation” ; “Fabrication of Durably Superhydrophobic Cotton Fabrics by Atmospheric Pressure Plasma Treatment with a Siloxane Precursor”; how is this article different, has superior or inferior properties, and why?

Answer: The innovation of our manuscript has been written in the last paragraph in Introduction. First, PMHS, one of organosilicone, is well-known for its very low surface energy. Second, PMHS possesses –Si-H bond which can react with –OH groups of cotton fabric at room temperature with Kastredt catalyst, therefore, the very hydrophobic PMHS chains can by grafted onto surface of cotton fabric easily. In our experiment, the bubbles of hydrogen gas were observed immediately as cotton fabric was immerged into the PMHS modifier solution. The detail discussion can refer to the Introduction of our manuscript.

  • Q2: In the title and in the abstract the second modification with fluorine olefin is not mentioned. There is the description in the experimental part and then in the results it is sometimes treated, sometimes not. The text must be revised so that all the characterizations are present and all the comments and descriptions in all the parts regarding this second modification.

Answer: OK, thank you very much. We have revised these in the new submitted manuscript.

Q3: Whenever an outcome is defined, an appropriate reference must be cited. For example at each attribution of the IR.

Answer: OK, thank you very much, we have cited references about IR, XPS in the new submitted manuscript.

Q4: Control of the English language used which sometimes needs to be improved for verbal forms and repetitions and check the references to figures that in some cases are missing or incorrect. Figure 1 (b) defined in the text where is it? The caption of figure 2 seems incorrect. Line 150, the references seem incorrect. Figures 3c and f, Figure 3 g, h, i are not commented on in the text.

Answer: OK, we have revised these. Thank you very much for your careful review on our manuscript. The XPS and SEM-EDS of fluorine olefin modified cotton fabric were commented in the Application in Stain-Resistance.

Q4: To verify the chemical reaction and also to evaluate its quantity, why have not been made thermogravimetry on the treated materials?

Answer: Thank you very much for your suggestion. Because of coronavirus, the testing center of our college is closed. So, we cannot carry out this experiment. However, we believe the characterization combined with the experimental phenomenon, the chemical reaction between modifiers and cotton fabrics can be demonstrated. The dehydrogenation between –Si-H and –OH is well-known in organosilicon chemistry.

Q5: In the section 3.5 the second treatment must be presented and compared with the first one. Furthermore, probably another index concerning the performance of the separation could be introduced. For example the time to separate the two phases.

Answer: PMHS modified cotton fabric had good oil/water separation ability. It is not necessary to carry out a further fluorine olefin modification. PMHS modification is good for oil/water separation and stain-resistance to water-borne pollutants. However, PMHS modified cotton fabric possesses poor stain-resistance to oil. Therefore, in our manuscript, we try to present a further modification of PMHS modified cotton fabric and its possibility in oil resistance. The detail experiments will be carried out in our future investigation. Thank you very much.

Reviewer 3 Report

The manuscript by Lin and co-workers escribes the surface modification of cotton and the resulting stain-resistance and applicability in oil-water separation. The research is of interest to the readers of Polymers. There is substantial amount of data backing up the claims. The research is timely and could have a good potential impact. However, there are several major and minor points that must be addressed prior to further consideration to publish.

1, The figure showing the WCA and water uptake as a function of PMHS concentration reveals that a very low degree of surface modification significantly changes the fabric characteristics but further modification does not bring any benefit. Therefore, why test further 5 points at high concentrations instead of trying to minimize the PMHS concentration below 0.1%?

2, The limitations of the proposed methodology should be discussed in the manuscript. What are the substrates that can be used for this surface modification? What kind of oils and other non-polar compounds can be separated? Demonstrate how the methodology can be applied to different scenarios and that it is of interest to a broad audience. This will help to increase the potential impact of the publication.

3, The concentrations in lines 79-80 cannot be interpreted. Are those wt%, vol% or something else? The amount of hexane and catalyst are not given. The experimental section should be more detailed to allow correct interpretation of the results as well as reproducibility of the work.

4, The purity and/or grade for all chemicals, solvents and materials used in the study should be given under the materials section of the manuscript.

5, There are a wide range of materials used for oil-water separation and they should be briefly mentioned in the manuscript (DOIs cotton fabric 10.1021/acssuschemeng.9b01122; graphene 10.1016/j.memsci.2020.118007; waste brick 10.1039/C9GC04178H; foams 10.1021/acsanm.9b02303; biomimetics 10.1039/C9EN01140D).

6, What is the molecular weight of the PMHS?

7, In Figure 4 the water adsorption should be expressed in g g-1 (i.e. gram of water over gram of fabric) as the percentage currently given does not reveal much.

8, The conclusion section should summarize the main research findings in some quantitative statements. In its current form, this section is too vague.

9, A comparison table, showing the state-of-the-art stain-resistant hydrophobic materials, should be included in the manuscript. This is to highlight the novelty of the proposed material and demonstrate its performance in comparison with other materials.

Author Response

Response to Reviewer 3:

Q1: The figure showing the WCA and water uptake as a function of PMHS concentration reveals that a very low degree of surface modification significantly changes the fabric characteristics but further modification does not bring any benefit. Therefore, why test further 5 points at high concentrations instead of trying to minimize the PMHS concentration below 0.1%?

Answer: This is because the reaction between PMHS and cotton fabric is very fast. After immerging cotton fabric into the PMHS modifier solution, the bubbles of hydrogen gas were observed immediately. So, the surface of cotton fabric will immediately covered by PMHS chains during modification. We also test PMHS concentration of 0.05% in the manuscript. Thank you very much.

Q2: The limitations of the proposed methodology should be discussed in the manuscript. What are the substrates that can be used for this surface modification? What kind of oils and other non-polar compounds can be separated? Demonstrate how the methodology can be applied to different scenarios and that it is of interest to a broad audience. This will help to increase the potential impact of the publication.

Answer: The oil used in our experiment is the diesel. Actually, we have also tested the oil/water separation of hexane/water. However, due to high volatility of hexane, the separation efficiency is not correct. So, we applied the commercialized oil, diesel, to test the oil/water separation efficiency. The suggestion is very helpful. For this manuscript, we cannot apply this suggestion because our lab is still closed due to the coronavirus. This is very helpful to us, we will apply this suggestion in our future research. Thank you.

Q3: The concentrations in lines 79-80 cannot be interpreted. Are those wt%, vol% or something else? The amount of hexane and catalyst are not given. The experimental section should be more detailed to allow correct interpretation of the results as well as reproducibility of the work.

Answer: Thank you very much. We have revised this in the new submitted manuscript.

Q4: The purity and/or grade for all chemicals, solvents and materials used in the study should be given under the materials section of the manuscript.

Answer: OK, we have added the information in the new submitted manuscript.

Q5: There are a wide range of materials used for oil-water separation and they should be briefly mentioned in the manuscript (DOIs cotton fabric 10.1021/acssuschemeng.9b01122; graphene 10.1016/j.memsci.2020.118007; waste brick 10.1039/C9GC04178H; foams 10.1021/acsanm.9b02303; biomimetics 10.1039/C9EN01140D).

Answer: Thanks for your suggestion, we have cited three of these references in the new submitted manuscript.

Q6: What is the molecular weight of the PMHS?

Answer: The MW of PMHS is about 1600, we have added in the new submitted manuscript.

Q7: In Figure 4 the water adsorption should be expressed in g g-1 (i.e. gram of water over gram of fabric) as the percentage currently given does not reveal much.

Answer: The adsorption test was applied to reveal the hygroscopicity. We have defined in the new submitted manuscript. Thank you.

Q8: The conclusion section should summarize the main research findings in some quantitative statements. In its current form, this section is too vague.

Answer: OK, thanks for your suggestion.

Round 2

Reviewer 1 Report

The authors satisfactory replied to some comments, in other cases they simply removed the results that they are unable to discuss. Sufficient improvement can be acknowledged. For correctness, the procedure used to prepare the samples for ATR and XPS (see the authors' answer to Q4) must be added in the experimental part. Text editing is required (especially new parts added to the text, some sentences need to be rephrased)
Please correct typos and mistakes in the manuscript,
examples are: Kastredt must be Karstedt, also in schemes 1 and 2; fluorine olefin : fluorinated olefin or hydrofluoroolefin
silixane : siloxane
32 scan times : 32 scans (line 111)  

Author Response

Responds to Reviewer 1

Q1: The authors satisfactory replied to some comments, in other cases they simply removed the results that they are unable to discuss. Sufficient improvement can be acknowledged. For correctness, the procedure used to prepare the samples for ATR and XPS (see the authors' answer to Q4) must be added in the experimental part. Text editing is required (especially new parts added to the text, some sentences need to be rephrased)

Answer: OK, we have added the procedure used to prepare the samples for ATR and XPS, and the whole manuscript was editing carefully again. Thank you very much.  

Q2: Please correct typos and mistakes in the manuscript, examples are: Kastredt must be Karstedt, also in schemes 1 and 2; fluorine olefin : fluorinated olefin or hydrofluoroolefin
silixane : siloxane
32 scan times : 32 scans (line 111)

Answer: It is very kind for you to point out these for us. We have revised Karstredt to Karstedt, changed fluorine olefin to fluorinated olefin, changed silixane to siloxane, and 32 scan times to 32 scans. Thank you very much.

Reviewer 2 Report

I understand the problems of adding characterizations to the data already presented now

Author Response

Responds to Reviewer 2

Q1: I understand the problems of adding characterizations to the data already presented now.

Answer: Thank you very much.

Reviewer 3 Report

The authors have addressed the comments, and the manuscript significantly improved.

Author Response

Responds to Reviewer 3

The authors have addressed the comments, and the manuscript significantly improved.

Answer: Thank you very much.
